# Carbon Ion Irradiation Downregulates Notch Signaling in Glioma Cell Lines, Impacting Cell Migration and Spheroid Formation

**DOI:** 10.3390/cells11213354

**Published:** 2022-10-24

**Authors:** Vivek Kumar, Mohit Vashishta, Lin Kong, Jiade J. Lu, Xiaodong Wu, Bilikere S. Dwarakanath, Chandan Guha

**Affiliations:** 1R&D Department, Shanghai Proton and Heavy Ion Center (SPHIC), Shanghai 201321, China; 2Shanghai Key Laboratory of Radiation Oncology (20dz2261000), Shanghai 201321, China; 3Shanghai Engineering Research Center of Proton and Heavy Ion Radiation Therapy, Shanghai 201321, China; 4Department of Radiation Oncology, Shanghai Proton and Heavy Ion Center, Fudan University Cancer Hospital, Shanghai 201321, China; 5Central Research Facility, Sri Ramachandra University, Porur, Chennai 600116, India; 6Albert Einstein College of Medicine, The Bronx, NY 10467, USA

**Keywords:** Notch signaling, Carbon ion irradiation, X-ray, migration, spheroid, stemness

## Abstract

Photon-based radiotherapy upregulates Notch signaling in cancer, leading to the acquisition of the stem cell phenotype and induction of invasion/migration, which contributes to the development of resistance to therapy. However, the effect of carbon ion radiotherapy (CIRT) on Notch signaling in glioma and its impact on stemness and migration is not explored yet. Human glioma cell lines (LN229 and U251), stable Notch1 intracellular domain (N1ICD) overexpressing phenotype of LN229 cells, and Notch inhibitor resistant LN229 cells (LN229R) were irradiated with either photon (X-rays) or (carbon ion irradiation) CII, and expressions of Notch signaling components were accessed by RT-PCR, Western blotting, and enzymatic assays and flow cytometry. Spheroid forming ability, cell migration, and clonogenic assay were used to evaluate the effect of modulated Notch signaling by irradiation. Our results show that X-ray irradiation induced the expression of Notch signaling components such as Notch receptors, target genes, and ADAM17 activity, while CII reduced it in glioma cell lines. The differential modulation of ADAM17 activity by CII and X-rays affected the cell surface levels of NOTCH1 and NOTCH2 receptors, as they were reduced by X-ray irradiation but increased in response to CII. Functionally, CII reduced the spheroid formation and migration of glioma cells, possibly by downregulating the N1ICD, as stable overexpression of N1ICD rescued these inhibitory effects of CII. Moreover, LN229R that are less reliant on Notch signaling for their survival showed less response to CII. Therefore, downregulation of Notch signaling resulting in the suppression of stemness and impaired cell migration by CII seen here may reduce tumor regrowth and disease dissemination, in addition to the well-established cytotoxic effects.

## 1. Introduction

Radiotherapy is one of the most commonly used treatments for a wide range of malignancies [1]. However, radioresistance in the cancer cells continues to be a key obstacle to the success of radiotherapy [2]. A recently developed particle therapy, especially carbon ion radiotherapy (CIRT), has emerged as a new promising therapeutic approach to treat cancers. CIRT has numerous therapeutic advantages over conventional radiotherapy, which include physical advantages, such as spread-out Bragg peak, enhanced dose distribution, lateral focusing, dose verification, superior linear energy transfer, and also biological advantages, such as lesser dependency on oxygen, induction of complex DNA damage, and higher relative biological effectiveness (RBE), (lower oxygen enhancement ratio; OER) [3]. Moreover, the carbon ion beam offers ideal energy distribution around the Bragg peak, which induces a maximum ionizing effect in the pathological lesion and less damage to the surrounding normal tissue, leading to better tumor controllability with shorter treatment times and fewer side effects [4]. Due to these advantages, the CIRT has shown a moderate improvement in the therapeutic outcome of some radio-resistant tumors; however, not to an extent as predicted by pre-clinical studies [5]. The reason behind this could be the inadequate knowledge of the events occurring at the molecular level that are responsible for the outcome of the therapy. Therefore, a detailed understanding of the molecular mechanisms underlying the effects of carbon ion irradiation (CII) on radio-resistant tumors such as glioblastoma (GBM) and normal cells is urgently required. This may also facilitate the optimization of photon and carbon ion irradiation combination-based radiotherapy for the treatment of radio-resistant tumors.

The Notch is an evolutionarily conserved signaling pathway that is involved in the regulation of many cellular processes throughout life, including cell fate decision, stem cell maintenance, differentiation, and proliferation [6]. Notch mediates short-range cellular communication through interaction with ligands presented on neighboring cells [7]. In mammals, Notch signals through four Notch receptors (NOTCH 1–4) and five Ligands (JAGGED-1, -2, and DELTA-like-1, -3, and -4), which are all type I transmembrane proteins. The activation of Notch signaling is initiated when the Notch receptor binds to the Notch ligands, followed by the sequential proteolytic cleavages of the Notch receptor by a member of the disintegrin and metalloproteinases (ADAM) family and gamma-secretase, which eventually leads to the release and nuclear translocation of the intracellular domains of Notch receptors (NICDs) and subsequent activation of Notch-dependent transcription [8,9,10]. Due to its pleiotropic role in various cellular functions, the dysregulation of Notch signaling is often observed in many types of cancers, including glioma [11,12]. Moreover, the Notch receptors, ligands, and target genes are frequently overexpressed in glioma [13], and the role of Notch signaling in the maintenance of glioma stem cells is well established [14,15].

Conventional radiotherapy, which is the major therapeutic strategy for treating glioblastoma, often upregulates Notch signaling, especially in glioma stem cells [16]. Furthermore, it was recently shown that the Notch signaling pathway is enriched in the invasive tumor cells that constitute a phenotypically distinct and highly radio-resistant GBM subpopulation with poor prognosis [17]. All of these form the major contributing factors in radioresistance that limit the efficacy of conventional radiotherapy. In this regard, the inhibition of Notch signaling in GBM leads to a decrease in the stem cells, thereby lowering the chance for radiation-induced stemness, thus, impairing the tumorigenic capacity of these cells and enhancing the sensitivity to photon radiation [18]. Moreover, Notch inhibitors are being evaluated as an adjuvant to conventional radiotherapy in the clinical trials for gliomas [19]. However, the impact of carbon ion irradiation (CII) on Notch signaling in gliomas has not been investigated so far. Therefore, in the present study, we investigated the impact of CII on Notch signaling in glioma cell lines and its effect on migration and spheroid formation, a phenotype to measure the stemness in glioma cell lines that are important in determining the local tumor control, recurrence, and invasive behavior of gliomas. Our results show that, unlike X-ray, CII reduces Notch signaling and reduces the migration and spheroid formation in LN229 and U251 glioma cell lines. Overexpressing Notch intracellular domain in LN229 attenuated the effect of CII on the migration and spheroid formation, implicating the role of Notch signaling in CII-mediated reduction in cell migration or spheroid formation. Moreover, Notch inhibitor (DAPT) resistant cells derived from LN229 cells were found to be less responsive to CII when compared to its parental cells, thereby strengthening the role of Notch signaling in the glioma cell response to CII.

## 2. Materials and Methods

### 2.1. Cell Lines

Human glioblastoma cell line LN229 was purchased from American Type Culture Collection (ATCC). Other human glioblastoma cell lines U251 and U87 were purchased from Procell (Wuhan, China). These cell lines were cultured in Dulbecco’s modified Eagle’s medium (DMEM, Sigma, St. Louis, MO, USA) supplemented with 10% fetal bovine serum (FBS, Hyclone, Logan, UT, USA), 100 U/mL penicillin, and 100 μg/mL streptomycin, and incubated at 37 °C and 5% CO_2_. All cell lines were cultured for less than 6 months following resuscitation and were confirmed mycoplasma-free by using the PCR-based mycoplasma detection kit (cat#C0301S, Beyotime-Biotechnology, China).

### 2.2. Generation of Dox-Inducible N1ICD LN229 Stable Cell Line

LN229 cells were co-transfected with either pB-TAG-N1ICD Plasmid (Addgene, cat #130934) or PB-TAG-ERP2 Plasmid (Addgene, cat# #80479) in addition to the Super PiggyBac transposase plasmid (System Biosciences, PB210PA-1) at a 1:5 ratio using Lipofectamine 3000 (Invitrogen, cat# L3000015) as described by the manufacturer. Puromycin (Beyotime, cat# ST551-10mg) selection was started two days after transfection and continued for 7 days with a daily change with fresh media containing Puromycin (1.0 mg/mL). Puromycin-resistant cells were expanded and tested for N1ICD induction by Doxycycline (DOX) (Beyotime, cat#ST039A) by Western blot. The induction of N1ICD by DOX (50 ng/mL) was performed 24 h before irradiation with X-rays or CII.

### 2.3. X-ray and Carbon-Ion Irradiation

Irradiation of cell lines was performed in standard culture flasks or culture dishes. A 225 kVp X-ray (13.30 mA) beam filtered with 2 mm AI by a XRAD225 from PXI Precision irradiator (Ge Inspection Technologies Shimadzu, Japan) at a dose rate of 3.2 Gy/min ± 0.02 was used for X-ray irradiation. The dose for X-ray was 2 Gy and 4 Gy. Carbon ion irradiation with doses 2 Gy and 4 Gy was performed using a heavy ion synchrotron accelerator (Iontris, Siemens AG, Munich, Germany) (IONTRIS intensity modulated raster scan system) at SPHIC as described before [20]. Briefly, CII was delivered as a homogeneous extended Bragg peak with an energy of 333.82 MeV/u. The delivered doses at the cell layer were verified by using an advanced Markus chamber (TM34045, PTW, Germany). This chamber was calibrated by using TRS-398. The dose averaged LET at the cell layer was calculated by using in-house software. The dose averaged linear energy transfer; (LETd) was 56.37 keV/μm on the target. The irradiation was performed at room temperature. It has to be emphasized that the accelerator beam time was very limited, which restricted the number of independent experiments.

### 2.4. RNA Isolation and RT-PCR

Total RNA was isolated from cells at 6 h and 24 h after irradiation using TRIzol reagent (Beyotime Cat#R0016) following standard protocol. Briefly, the media was discarded, and 1 mL TRIzol reagents were added and were pipetted vigorously. It was incubated for 5 min at room temperature before adding 0.2 mL chloroform. Tubes were vortexed for 15 s and incubated at room temperature for 2–3 min, followed by centrifugation at 12,000× *g* for 15 min at 4 °C. The aqueous phase was transferred to a fresh tube, and 0.5 mL isopropanol was added. Samples were incubated at room temperature for 10 min. Following centrifugation at 12,000× *g* for 10 min at 4 °C, the RNA pellet was washed once with 1 mL 75% ethanol. Samples were vortexed and centrifuged at 7500× *g* for 5 min at 4 °C. The RNA pellet was air-dried for 5–10 min and dissolved in 25–30 μl RNase-free water. RNA concentration was determined by Nanodrop (Biorad), and cDNA was prepared from 1000 ng RNA using the PrimeScript™ RT reagent Kit with gDNA Eraser (Perfect Real Time) (Cat# RR047A; TAKARA) following the manufacturer’s protocol. The product was diluted in RNase-free water, and 25 ng was used for quantitative RT-PCR using SYBR^®^ Premix Ex Taq™ (Tli RNaseH Plus) (Cat # RR420A; TAKARA). ACTB was used as an internal control, and 2^−ΔΔCt^ was used to calculate gene expression. The following primer sequences were used for the real-time PCR:

*Notch1*-F: 5′-GGTGAGACCTGCCTGAATG-3′

*Notch1*-Rv: 5′-GTTGGGGTCCTGGCATC-3′

*Notch2*-F: 5′-TGGGCTACACTGGGAAAAAC-3′

*Notch2*-Rv: 5’-ACATAGGCACTGGGACTCTG-3′

*Notch3*-F: 5′-CGTGGCTTCTTTCTACTGTGC-3′

*Notch3*-Rv: 5′-CGTTCACCGGATTTGTGTCAC-3′

*Hes1*-F: 5′-AGTGAAGCACCTCCGGAAC-3′

*Hes1*-Rv: 5′-TCACCTCGTTCATGCACTC-3′

*Hey1*-F: 5′-ATCTGCTAAGCTAGAAAAAGCCG-3′

*Hey1*-Rv: 5′-GTGCGCGTCAAAGTAACCT-3′

*Dll1*-F: 5′-GATGTGATGAGCAGCATGGA-3′

*Dll1*-Rv: 5′-CCATGGAGACAGCCTGGATA-3′

*Dll4*-F: 5′-TCCAACTGCCCTTCAATTTCAC-3′

*Dll4*-Rv: 5′-CTGGATGGCGATCTTGCTGA-3′

*Jag1*-F: 5′-GAATGGCAACAAAACTTGCATG-3′

*Jag1*-Rv: 5′-AGCCTTGTCGGCAAATAGC-3′

*Jag2*-F: 5′-ATGAGTGTGAAGGGAAGCCA-3′

*Jag2*-Rv: 5′-GTCGTTGACGTTGATATGGCA-3′

*Actb*-F: 5′-CACCAACTGGGACGACAT-3′

*Actb*-Rv: 5′-ACAGCCTGGATAGCAACG-3′

### 2.5. Western Blotting

After irradiation, the cells were washed with cold PBS and incubated with RIPA lysis buffer (EMD Millipore Corp., Cat#20-188) supplemented with a complete protease inhibitor cocktail and phosphatase inhibitors (Thermo Scientific, Cat#A32955) for 10 min on ice, and thereafter, sonicated with 3 pulses at 50% amplitude with a sonicator (Sonic, VibraCell, Sonic&Materials Inc., Newton, CT, USA). The cell lysates were centrifuged at 12,000 rpm for 20 min at 4 °C, and supernatant was collected. Protein concentration was determined using the Bradford reagent (Beyotime, Cat#P0006C). An equal amount of proteins from each sample (45 µg) was denatured by heating at 95 °C for 5 min with Laemmli loading buffer and loaded onto a 8% SDS-polyacrylamide gel. After separation, proteins were transferred onto PVDF membranes (Amersham Hybond, Cat#10600029). The membranes were blocked in 5% BSA in TBST [25 mM Tris·HCl (pH 7.5), 125 mM NaCl, and 0.1% Tween 20] for 1 h and incubated with the indicated primary antibodies: NOTCH1 (Cell Signaling, Cat#3608S), NOTCH1Val1744 (Cell Signaling, Cat#4147S), NOTCH2 (Cell Signaling, Cat#4530S), and β-ACTIN (Santa Cruz Biotechnology Inc., Cat#SC-47778) overnight at 4 °C. Membranes were washed three times with TBST for 10 min each and incubated with horseradish peroxidase-conjugated secondary antibodies for 1 h at room temperature. The immune complexes were detected by chemiluminescence (BioRad Clarity Western ECL Substrate, Cat#170-5060), and images were recorded with the help of chemidoc (Biorad). Densitometric analyses of the band intensities in the blot were performed using Image Lab software (version 6.1.0 build7 standard edition, Bio-Rad Laboratories).

### 2.6. Cell Surface Notch1 and Notch2 Staining and Analysis by Flow Cytometry

At 6 h and 24 h after irradiation, cells were dislodged by pipetting. Cells were fixed in 4% PFA in HBSS containing cation, washed thrice with HBSS containing cation, and stored at 4 °C until use. Fixed cells were washed once with 1 mL Ligand Binding Buffer (LBB) (HBSS + 1% BSA + 1 mM CaCl_2_ pH 7.2–7.4), and FcR was blocked with Human TruStain FcX (BioLegend Cat#422302) in 90 uL LBB. Following incubation on ice for 15 min, 10 uL mix of anti-NOTCH1-PE Ab (Biolegend, Cat#352106, Clone;MHN1-519) and anti-NOTCH2-APC Ab (Biolegend, Cat# 34806, Clone:MHN2-25) was added. After incubation for 30 min at 4 °C in the dark, the cells were washed twice with 1 mL LBB, and Notch1 and Notch2 surface expression was determined using a Cytoflex S flow cytometer (Beckman Coulter), and data were analyzed using FlowJo Software (version 10.4, FlowJo, LLC).

### 2.7. ADAM17 Activity Analysis

Following the manufacturer’s instructions, ADAM17 activity was assessed using the SensoLyte 520 TACE (α-Secretase) Activity Assay Kit (ANASPEC, Cat#AS-72085). Briefly, 6 h or 24 h after irradiation, cells were washed with PBS, collected, and lysed with assay buffer. TACE substrate was added to the lysate, and after 45 min, ADAM17 enzymatic activity was quantified by measurement of fluorescence intensity in a microplate fluorometer (λ_ex_ 485 nm and λ_em_ 535 nm). ADAM17 activity was normalized with the total protein level of each sample as determined by the Bradford protein assay (Beyotime, Cat#P0006C).

### 2.8. Scratch Assay

A scratch assay was used to access the migratory property of the cells after X-rays or carbon ion irradiation. Briefly, after irradiation, approximately 0.2 × 10^6^ cells were seeded in 24-well plates in complete DMEM and allowed to form a confluent monolayer for 10–12 h in CO_2_ incubator. A scratch was made using a sterile 1000 uL pipette tip, followed by washing the cells with sterile PBS to eliminate the dislodged or non-adherent cells. The DMEM media containing 1% FBS was added to the cells. Microphotographs were taken at 0 h and 24 h under light microscope after scratching. The percentage of migration of irradiated cells was quantitated by measuring the area of the cell-free zone immediately after making the scratch at 0 h and 24 h later using image analysis software (version 1.53a, Image J2). Changes in migration potential of the irradiated cells were expressed as relative fold changes to the unirradiated controls.

### 2.9. Colony Formation Assay

The cell survival after irradiation was assessed by the colony formation assay. Cells were washed with 1X PBS after CII irradiation, and the single cell suspension was prepared by trypsinization. The cells were counted and seeded in duplicate or triplicate at appropriate cell densities for colony formation in 6-well plates. The plated cells were cultured for 9–12 days at 37 °C in humidified 5% CO_2_ atmosphere. Colonies were fixed in PBS: methanol (1:1) and stained with 1% crystal violet. Colonies containing more than 50 cells were counted. Images of each group were captured by a colony counting machine (Gel-Count, Oxford Optronix Ltd., Milton Park, Oxford, OX14 4SA, UK). Surviving fractions were calculated based on the plating efficiencies of the unirradiated cells.

### 2.10. Generation of Notch Inhibitor (DAPT) Resistant LN229 (LN229R) Glioma Cell Lines

The Notch Inhibitor (DAPT)—resistant LN229 glioma cell line was derived from the original parental cell line by continuous exposure to stepwise increasing concentrations of Notch Inhibitor (DAPT). Initially, the exponentially growing parental LN229 cells were exposed to low-dose (1 uM) difluorophenyl)acetamido]propanamido}phenylacetate (DAPT) (Millipore, Cat#565784), which is a gamma-secretase inhibitor (GSI) that inhibits Notch signaling. These cells were maintained in DAPT containing DMEM medium supplemented with 10% FBS and 1% penicillin/streptomycin and subcultured upon reaching 70–80% confluency for 1 week. At this point, the concentration was increased 5-fold, and the above process was repeated until the final concentration reached 125 uM. The DAPT-resistant clones (LN229R) were selected and cryopreserved. The resistivity of LN229R to DAPT was confirmed by a cell viability assay.

### 2.11. Cell Viability Assay

The cell viability assay was performed on each sample according to the manufacturer’s instructions (Cell Titer Lumi, Beyotime, Cat#C0065L). Briefly, 4000 cells were plated in duplicate onto 96-well plates. After incubation overnight, cells were treated with the indicated concentration of DAPT (Millipore, Cat#565784) and incubated at 37 °C in CO_2_ for 72 h. Then, 100 uL of Cell Titre Lumi was added into each well and incubated for 10 min, followed by transfer of the mixer into the white 96-well plate. The luminescence was measured using a 96-well plate reader (Biotek). IC50 values were calculated using GraphPad Prism (version 8.4.0).

### 2.12. Spheroid Formation Assay

After irradiation, the cells were detached with 0.05% trypsin/EDTA (Gibco, Germany), and after trypsin inactivation, the resulting single cells were washed twice with PBS. Then, 40,000 or 0.1 × 10^6^ cells were incubated in 500 uL or 2 mL spheroid forming media (DMEM/F12, Gibco, Germany), which was supplemented with 20 ng/mL epidermal growth factor (EGF, PeproTech, Cranbury, NJ, USA), 10 ng/mL of basic fibroblast growth factor (bFGF, PeproTech, USA), 2% B27 supplement (Gibco, Germany), 1% penicillin–streptomycin and seeded in low attachment dish or poly-Hema coated 24-well plates for 10–14 days. The media was changed every 3rd or 4th day to fresh spheroid forming media. Spheroids formed were counted and captured with an inverted microscope. Only the spheroid with a size greater than 100 um was counted.

### 2.13. Cell Growth Inhibition Analysis

At 24 h prior to irradiation, cells were treated with either DMSO or DAPT (10 uM) Cells were irradiated with either CII (2Gy) or X-ray (2Gy), and an equal number of cells were seeded in 24-well plate. Cells were harvested at 48 h after irradiation by trypsinization, and the cell number was enumerated with the help of a cell hemocytometer. The extent of cell growth inhibition by treatment was assessed by calculating the cell growth as follows:Cell growth at 48 h = N_t_/N_0_;
Relative cell growth (%) = [Cell growth (Treatment)/Cell growth (DMSO)] × 100
where N_t_ is the cell number at 48 h post irradiation and N_0_ is the number at the time of seeding.

### 2.14. Statistical Analysis

All analytical data are presented as means ± SD. The significance was determined by a one-way or two-way ANOVA, followed by Bonferroni post-hoc test, where multiple groups were compared, and two-tailed, unpaired, parametric, Student’s t-test using GraphPad Prism software. The significance levels are indicated in the legends of the figures.

## 3. Results

### 3.1. In Contrast to X-rays, CII Reduces the Expression of Genes Involved in Notch signaling in Glioma Cell Line

Photon irradiation was demonstrated to upregulate Notch signaling in cancer, which was linked to the therapy induced development of radioresistance [16]. To investigate the effect of CII on Notch signaling, we analyzed the dose-dependent changes in the mRNA levels of Notch receptors (*Notch1*, *Notch2*, and *Notch3*) and target genes (*Hes1* and *Hey1*) in LN229 and U251 cells and compared them with the changes following X-ray irradiation in these two cell lines. In LN229 cells, X-rays upregulated the mRNA expression of *Notch1* and *Hey1* genes in a dose-dependent manner observable at 24 h post irradiation, whereas no significant modulation was detectable for other Notch receptors and targets (Figure 1A,B). Interestingly, with CII, no significant upregulation of either Notch receptors or targets was observed (Figure 1A,B). Moreover, *Hes1* mRNA was reduced at 24 h following 2 Gy CII (Figure 1A), while the levels of *Hes1*, *Notch2*, and *Hey1* were reduced at 6 h following 2 Gy and 4 Gy CII (Appendix A). In U251, X-ray induced the mRNA expression of *Notch1*, *Notch2,* and *Notch3* receptors in a dose-dependent manner, while the targets genes were not affected at 6 h (Appendix A). At 24 h after X-ray irradiation, only *Notch2* and *Hes1* were upregulated (Figure 1C,D). Similar to LN229, CII either had no effect on the Notch receptor and target gene expression or reduced the *Notch1* and *Hey1* at 6 h (Appendix A) or *Hes1* and *Hey1* at 24 h (Figure 1C,D) in U251 cells. These differential effects of CII and Xray on the mRNA expression of Notch receptors and targets genes were also observed in U87 cells lines (Appendix A). At 6 h after irradiation, only *Notch2* and *Hey1* were significantly reduced by 4Gy CII, while other Notch receptors or target genes were not modulated by either CII or X-ray (Appendix A). However at 24 h, 4Gy X-rays significantly induced the expression of Notch receptors (*Notch1*, *Notch2*, and *Notch3*) and target genes (*Hes1* and *Hey1*), while 4Gy CII reduced *Notch2* and *Hey1* (Appendix A). In LN229 and U251 cells, we also examined the mRNA expression of Notch ligands after CII or X-ray irradiation at 6 h or 24 h. While CII did not modulate any Notch ligands in LN229 or U251 at 6 h or 24 h (Appendix A), X-rays at 2 Gy induced *Dll4* at 6 h and *Dll1* at 24 h in LN229 cells (Appendix A), while a significant induction of *Jag1* mRNA expression was noted with 4 Gy X-rays at 6 h and 24 h in LN229 cells (Appendix A). In U251, only 4Gy X-rays induced the *Dll1, Dll4*, and *Jag1* mRNA expression at 24 h (Appendix A), while at other doses and time points, no significant modulation of Notch ligands was observed in U251 (Appendix A–C). As Notch1 and Notch2 play a major role in the glioma tumorigenesis and in therapy resistance, we next examined the effects of X-rays and CII on the protein expression of NOTCH1 and NOTCH2 receptors. An increase in the protein levels of NOTCH1 intracellular domain (N1ICD) was observed at 24 h following X-rays in both LN229 and U251 cell lines (Figure 2B,D), while NOTCH2 intracellular domain (N2ICD) was increased only in U251 (Figure 2D). Moreover, an increase in the protein levels of activated NOTCH1 as detected by antibody against NOTCH1Val1744 was observed at 24 h following X-rays in both LN229 and U251 cell lines (Appendix A). Interestingly, the N1ICD protein level and activated NOTCH1 were reduced by CII at 24 h in a dose-dependent manner in both the cell lines (Figure 2A,C and Appendix A). The N2ICD protein level was reduced in U251 by CII (Figure 2C), while it remained unaltered in LN229 cells (Figure 2A). In U87, a similar effect of CII and X-ray was observed on the protein expression of N1ICD (Appendix A). These results suggest that X-ray and CII differentially regulate the mRNA expression and protein levels of the Notch receptor and target gene in these glioma cell lines.

### 3.2. Differential Effect of X-ray and CII on ADAM17 Activity and Surface Notch Receptors Level

ADAM17 (a disintegrin and metalloproteinase) is a membrane-associated metalloproteinase that is involved in the S2 cleavage of Notch receptors at the cell surface [21], thereby affecting the surface level of Notch receptors. Moreover, X-ray increases the ADAM17 activity in non-small cell lung cancer in a time and dose-dependent manner, thereby enhancing the cleavage of its substrates on the cell surface [22]. Therefore, we investigated to see if the reduction in the NOTCH1 and NOTCH2 intracellular domain or Notch1Val caused by CII is due to an effect on the ADAM17 activity. In line with the earlier report, we also found that X-ray induced the ADAM17 activity at 24 h in both the cell lines (Figure 3A,D), although at 6 h this effect was not significant (Appendix A). Interestingly, CII significantly reduced the ADAM17 activity in both LN229 and U251 cells at 6 h (Appendix A) and 24 h (Figure 3A,D). We reasoned that because ADAM17 is involved in the S2 cleavage of Notch receptors at the cell surface, the differential regulation of ADAM17 activity by CII and X-rays would affect the availability or the level of Notch receptors on the cell surface. To verify this, we determined the surface level of NOTCH1 and NOTCH2 receptors after CII or X-ray irradiation. At 6 h after CII irradiation, no significant change in the surface level of the NOTCH1 receptor was observed in LN229 (Appendix A), while in U251, CII increased the surface level of NOTCH1 (Appendix A). X-rays did not cause any change in the surface level of NOTCH1 at 6 h in both the cell lines (Appendix A). At 6 h, 4 Gy CII had no significant impact on the surface level of NOTCH2 in LN229 cells, while an increase was noted in U251 (Appendix A). Similar to NOTCH1, the surface level of NOTCH2 was not affected by X-rays at 6 h (Appendix A). These observations suggest that the cleavage of surface Notch receptor at an early time after irradiation (6 h) varies in these two cell lines. At 24 h after irradiation, CII significantly increased the cell surface level of NOTCH1 in LN229 cells, which was in contrast to the levels seen after X-rays, where a decrease was evident (Figure 3B). Similar differential changes in the surface levels of NOTCH2 were observed in LN229 following CII and X-rays (Figure 3C). Interestingly, in U251 cells, X-rays did not affect the surface level of either NOTCH1 or NOTCH2, while there was a trend towards an increased surface level of either NOTCH1 following CII, although it was not significant (Figure 3E). However, 4Gy CII significantly increased the surface level of NOTCH2 in U251 cells (Figure 3F).

### 3.3. CII Attenuates the Migration and Spheroid Forming Ability of LN229 and U251 Glioma Cells in Contrast to an Induction by X-rays

Radiation-induced migration and acquisition of the stem phenotype are still the major contributing factors that are involved in the development of radioresistance and tumor resurgence after radiotherapy [23]. Moreover, these two cellular events are well known to be regulated by Notch signaling [6]. Therefore, to investigate the functional impact of the CII-mediated reduction in NICD, we examined radiation-induced migration and acquisition of the stem phenotype in glioma cell lines. We used the spheroid forming ability of the glioma cell line post irradiation to access the radiation-induced acquisition of stemness because it is one of the most dependable and proven parameters to assess the stemness of the cancer cells. Interestingly, CII reduced the migration of the LN229 cell line (Figure 4A,B), whereas X-ray promoted the LN229 cell migration at both doses (Figure 4A,C). A similar effect on the migration of U251 cell lines was observed with CII (Figure 4D,E) and X-rays (Figure 4D,F). CII also affected the spheroid forming ability of LN229 and U251 cell lines post irradiation. As shown in Figure 5A,B, CII significantly reduced the spheroid formation of LN229 and U251, while this effect was not observed with the X-ray. These findings imply that CII and X-rays have differential effects on migration and spheroid formation in glioma cell lines, thus, matching the differential impact of CII and X-ray on Notch signaling.

### 3.4. Overexpression of N1ICD Blunts the Effect of CII on Migration and Spheroid Formation in LN229 Cells

The *Notch1* gene is reported to be significantly higher in the glioma stem cells and in the neurosphere, than in monolayer cultures [24], and *Notch1* is highly correlated with the stemness marker among GBM cell lines, suggesting its role in glioma stem maintenance [24]. Moreover, the role of NOTCH1 in glioma cell migration and invasion is well established [24,25]. Therefore, to investigate the effect of exogenous overexpression of NOTCH1 intracellular domain (N1ICD) on CII-mediated repression of migration and spheroid formation, we established the stable Doxycycline (Dox)-inducible NOTCH1 intracellular domain expressing LN229 cells. This leads to the Dox inducible overexpression of N1ICD, thereby bypassing the proteinase cleavage event, which was downregulated by CII (Figure 2A). The overexpression of N1ICD in LN229 glioma cells was confirmed by the Western blot (Appendix A). The overexpression of NICD increased the spheroid forming ability (Appendix A) and migration potential of LN229 cells (Appendix A). Next, we reasoned if Notch signaling is involved in CII-mediated inhibition in migration and spheroid formation, then the overexpression of N1ICD in LN229 would offset the inhibitory effects of CII on the migration and spheroid formation. Indeed, as shown in Figure 6B, the overexpression of N1ICD in LN229 attenuated the inhibitory effects of CII on migration (Figure 6A). Similarly, the inhibitory effect of CII on the spheroid forming ability of LN229 was also reduced when N1ICD was overexpressed (Figure 6C). These results suggest that CII partially mediates its effect on LN229 glioma cell migration and spheroid formation via Notch signaling.

### 3.5. Notch Inhibitor Resistant Glioma Cells Are Less Responsive to CII-mediated Effects

To investigate if the glioma cell line’s reliance on Notch signaling affects its response to CII, we established a Notch inhibitor resistant glioma cell line using continuous exposure of LN229 to stepwise increasing concentrations of Notch signaling inhibitor DAPT (Gamma Secretase Inhibitor, involved in the S3 cleavage of Notch receptors) and selecting the resistant clone (LN229R). The cell viability assay revealed that LN229R cell lines were more resistant to DAPT, with an IC50 value of 153 uM at 72 h, than its parental LN229 cell lines, which had an IC50 value of 34.39 uM (Figure 7A), indicating that LN229R cells are less dependent on Notch signaling for survival. Interestingly, the clonogenic survival assay following CII showed that LN229R cell lines were resistant to CII, as they demonstrated enhanced clonogenic survival after CII when compared to the parental LN229 cell lines (Figure 7B). The LN229R cell was also less responsive to CII-mediated inhibition of migration, as 2 Gy CII could not inhibit the migration of LN229R (Figure 7C) while it effectively inhibited the migration in LN229 at the same dose (Figure 4B). Although at a higher dose of 4 Gy, the CII inhibited the migration in LN229R (Figure 7C). Similarly, the LN229R cells were less susceptible to CII-mediated inhibition of spheroid formation when compared to its parental LN229 cell lines at 2 Gy, but at a higher dose of 4 Gy, CII effectively reduced the spheroid formation in LN229R similar to its parental counterpart (Figure 7D).

## 4. Discussion

Radiation-induced Notch signaling in cancers is regarded as one of the contributing factors involved in the acquisition of resistance to the therapy associated with the induction of stem phenotype and invasive ability, which contributes to the recurrence and metastasis. Carbon ion radiotherapy (CIRT) has emerged as a superior radiotherapy modality over photon-based therapies due to its physical, ballistic, and biological advantages, while the molecular mechanism underlying the biological response of CII, especially in a radio-resistant cancer such as glioma, is not yet fully understood. Results of the present study show that in contrast to X-rays, CII either does not induce or downregulate Notch signaling by reducing the transcription (mRNA) and protein levels of important regulators of Notch signaling in human glioma cell lines, which correlated with CII-mediated reduction in the spheroidogenesis and migratory potential, supporting the notion that Notch signaling is involved in CII-mediated inhibition of both these responses. In addition, our results also suggest that, unlike X-rays, CII attenuates the activity of ADAM17, one of the metalloproteinases involved in the S2 cleavage of Notch receptors. Although the earlier study has shown that CII and photon irradiation elicit distinct cellular transcriptome, proteosome, and phosphatome responses in human lung adenocarcinoma cells [26], this is the first study showing the differential modulation of key Notch signaling regulators by X-rays and CII, as well as the relationship between CII-mediated downregulation of the Notch signaling and reduced spheroidogenesis and migratory potential following CII. The results of our study also complement a recent report, where the transcriptome analysis of the in vivo glioma tumor grafted in syngenic mice irradiated with CII was found to downregulate genes involved in Notch signaling [27]. Furthermore, CII was found to downregulate the regulators of other signaling pathways that directly or indirectly regulate Notch signaling components, offering a potential explanation for the CII-mediated downregulation of Notch signaling seen in our study [27]. Similarly, the integrin pathway, which is well known to regulate Notch signaling, was also shown to be downregulated by CII in glioma [28]. However, a recent study on cervical cancer showed upregulation of Notch signaling by CII [29]. These contrasting observations suggest that CII-mediated effect on Notch signaling could be dependent on the cancer type, which needs to be explored further.

Cell-membrane-associated metalloproteinase ADAM17 and ADAM10 cooperate with gamma-secretase to sequentially cleave the Notch receptors, resulting in the generation of the intracellular domain of Notch receptors (NICDs), which leads to the activation of Notch signaling [7]. X-rays were shown to induce ADAM17 activity but not ADAM10 activity without affecting their gene expression [22], while the effect of CII on ADAM17 activity has not been investigated so far. Downregulation of ADAM17 activity by CII, found in our study, is in-line with the reduced levels of N1ICD, N2ICD (Figure 2), and activated NOTCH1 (Appendix A), as well as the increase in the surface level of N1 and N2 observed following CII (Figure 3). On the contrary, an upregulation in ADAM17 activity was observed following X-rays, as reported in the earlier studies [22]. Taken together, these observations suggest a role of ADAM17 in the regulation of N1ICD, N2ICD activated Notch1 protein levels post-CII (Figure 2 and Appendix A). Since oxidative stress is correlated with hyperactivation of the ADAM17/Notch signaling pathway in fibrosis [30], the differential regulation of ADAM17 activity by X-rays and CII could be due to dissimilar spatiotemporal generation of ROS by X-rays and CII, thereby affecting the oxidative stress response. ROS generated by X-rays is more diffused and elicits a strong oxidative stress, while the ROS generated by CII is more localized and elicits a weak oxidative stress [31]. Additionally, this dissimilar spatiotemporal generation of ROS by X-rays and CII could influence the critical regulators of oxidative stress such as NRF2, which are known to regulate many of the Notch signaling components [32]. It would also be interesting to see if CII alters the gamma-secretase activity involved in the S3 cleavage of the Notch receptor.

The Notch signaling is known to play a crucial role in the regulation of migration, invasion, differentiation, survival, and maintenance of cancer stem cells population in various tumors, including glioma [6,33,34,35]. Moreover, its upregulation by photon irradiation was also linked to induced migration and acquisition of stemness, leading to radioresistance and failure of therapy [16,18,36]. In gliomas, photon irradiation is reported to significantly enhance migration, invasion [37], and stemness [38], while CII was found to reduce the migration [28], invasion [17] and stemness [27]. Our results are in line with these observations and implicate a role for Notch signaling in the differential response between photon irradiation and CII. Overexpression of N1ICD countered the effect of CII in reducing migration (Figure 6B) and spheroid formation (Figure 6C) and also enhanced the spheroid frequency in the unirradiated cells (Appendix A), lending support to the notion that compromised Notch signaling by CII could be responsible for reduced migration and spheroid formation. Although, the migration assay clearly demonstrates the significance of Notch signaling in CII-mediated effect on cell migration, it would be worthwhile to examine the molecular mechanism underlying this phenomenon, such as the EMT pathway where Notch signaling plays a crucial role [39], to further strengthen our observation. Similarly, apart from the spheroid forming assay utilized here, further investigations using markers of stemness in glioma are expected to enhance our understanding of CII-mediated suppression of Notch signaling and its effect on the stemness of glioma.

A reduction in the CII-induced cell death (Figure 7B; clonogenic survival), enhanced migration and spheroid formation in LN229R cells that are less dependent on functional Notch signaling compared to the parental cell line (LN229) suggests that CII could be less effective in the cells that acquire resistance to Notch inhibitor or are less dependent on Notch signaling for their survival and functioning. Although, at the higher dose (4Gy), CII was effective in reducing the migration and spheroid formation in LN229R, it could be due to the incomplete acquisition of resistance to the Notch inhibitor in LN229R. Therefore, it would be worthwhile to investigate the effects of CII on glioma cells that are more resistant to Notch inhibitors. This may also help in stratifying patients who can benefit from CIRT based on the status and dependency of tumors on the Notch signaling they harbor. The results of our investigations also point out the advantages of CII over the use of Notch signaling inhibitors such as DAPT. In clinical studies for gliomas, Notch inhibitors are now being assessed as an adjuvant to radiotherapy [19], although the outcome of this trial is not satisfactory so far. This could be due to the systemic inhibition of Notch signaling by these drugs, which often leads to negative side effects. Additionally, the dose of the Notch inhibitor required to block photon-induced Notch signaling frequently has an off-target effect that makes it difficult to include in the conventional treatment regimens. In this context, due to the physical characteristics of CII, the targeted inhibition of Notch signaling by CII in the tumor may be advantageous when compared to the systemic inhibition of Notch signaling by Notch inhibitors. Moreover, the use of a Notch inhibitor such as DAPT with CII could have beneficial effects as the combination of DAPT and CII significantly inhibited the cell growth when compared to either DAPT or CII alone (Appendix A). Although very primitive, the results of the present studies also suggest a potential benefit of priming with CIRT before applying photon-based RT while planning in the photon-CIRT combinational therapy [40] for GBM, as prior treatment with photons may upregulate Notch signaling and compromise the clinical efficacy of CIRT.

## 5. Conclusions

Taken together, the results of the present study show that in contrast to X-rays, CII very effectively downregulates Notch signaling, which appears to have an impact on the inhibition of migration and spheroid formation, two important factors that contribute to the failure of radiotherapy. Reduced suppressive effects of CII on the migration and spheroid formation by overexpression of N1ICD and relative refractoriness to CII in cells that have acquired resistance to Notch signaling inhibition by DAPT provide support to this proposition. Whether these in vitro observations will translate into improved clinical outcomes remains to be investigated using appropriate pre-clinical in vivo models.

## Figures and Tables

**Figure 1 cells-11-03354-f001:**
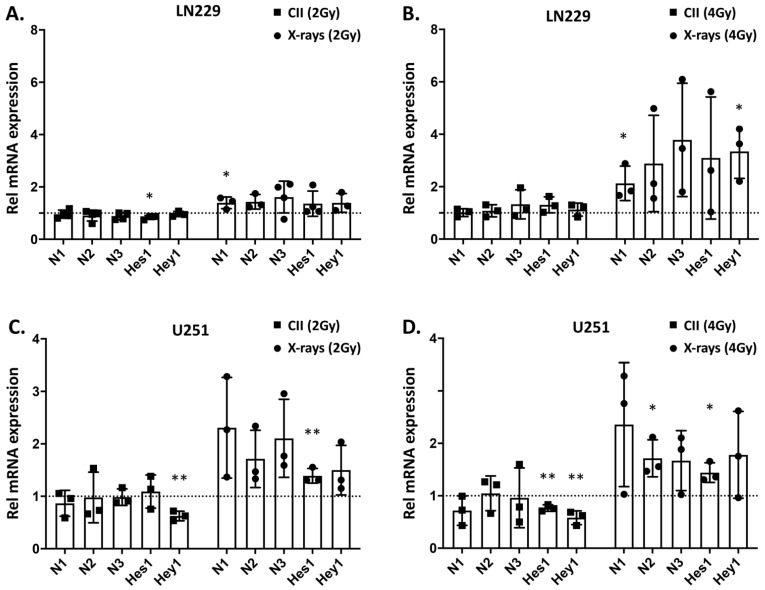
Differential regulation of Notch signaling genes by CII and X-rays in human glioma cell lines: The mRNA expression of Notch receptors (relative to un-irradiated sample): *Notch1* (N1), *Notch2* (N2), and *Notch3* (N3); Notch target genes: *Hes1* and *Hey1*, observed at 24 h after irradiation were analyzed by qRT-PCR in LN229 and U251 monolayer cell culture irradiated with either (**A**,**C**) 2 Gy X-ray or carbon ion irradiation or; (**B**,**D**) 4 Gy X-ray or carbon ion irradiation. Each symbol represents an independent experiment performed in triplicate. Data are presented as mean ± SD. *p* values were determined by an unpaired two-tailed Student’s *t*-test, * *p* < 0.05, ** *p* < 0.01.

**Figure 2 cells-11-03354-f002:**
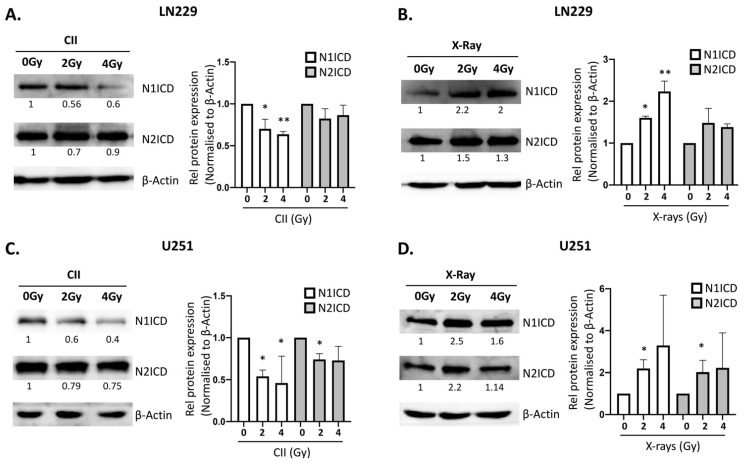
CII reduces while X-Ray enhances NOTCH1ICD and NOTCH2ICD protein levels: Western blot analysis of NOTCH1, NOTCH2, and β-ACTIN as a loading control in (**A**) LN229 irradiated with CII, (right) densitometry analysis of protein level in panel A; values are average of three independent experiments; (**B**) LN229 irradiated with X-Ray, (right) densitometry analysis of protein level in panel B, values are average of two or more independent experiments; (**C**) U251 irradiated with CII, (right) densitometry analysis of protein level in panel C, values are average of three independent experiments; (**D**) U251 irradiated with X-Ray, (right) densitometry analysis of protein level in panel D, values are average of two or more independent experiments. Data are presented as mean ± SD. Statistical analysis was performed by ANOVA with Bonferroni post-hoc test. *p*-value (* *p* < 0.05, ** *p* < 0.01).

**Figure 3 cells-11-03354-f003:**
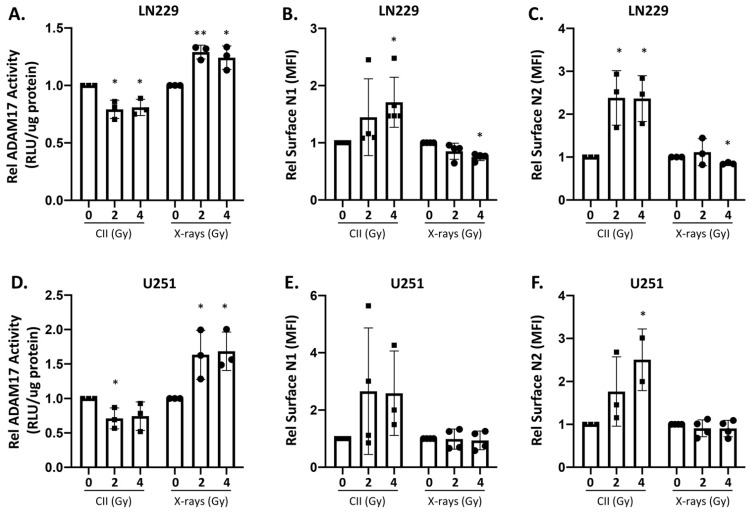
Differential regulation of ADAM17 activity and cell surface Notch receptors by CII and X-rays in human glioma cell lines: Relative ADAM17 activity at 24 h after irradiation with CII and X-rays in (**A**) LN229 and (**D**) U251 cells. Relative MFI of cell surface NOTCH1 receptor (N1) in (**B**) LN229 and (**E**) U251 and NOTCH2 receptor (N2) in (**C**) LN229 and (**F**) U251, as determined by flow cytometry at 24 h after irradiation with CII and X-rays. Each symbol represents an independent experiment. Data are presented as mean ± SD. Statistical analysis was performed by ANOVA with Bonferroni post-hoc test. *p*-values (* *p* < 0.05, ** *p* < 0.01).

**Figure 4 cells-11-03354-f004:**
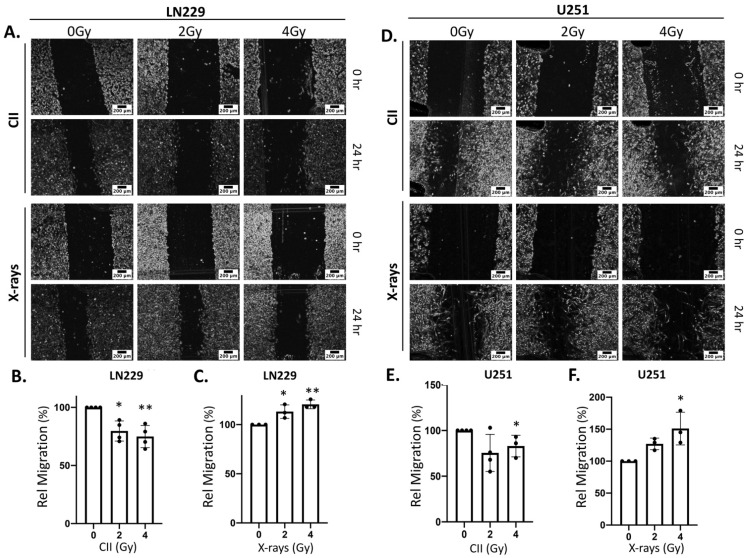
CII reduces while X-rays enhances migration of LN229 and U251 glioma cells: Representative images of the (**A**) LN229 and (**D**) U251 glioma cell migration after irradiation with CII or X-rays as analyzed by scratch assay (scale bar in black = 200 μm); (**B**) Bar graph showing the relative migration of LN229 irradiated with CII as derived from panel (**A**); (**C**) Bar graph showing the relative migration of LN229 cell irradiated with X-rays as derived from panel (**A**); (**E**) Bar graph showing the relative migration of U251 cell irradiated with CII as derived from panel (**D**); (**F**) Bar graph showing the relative migration of U251 cell irradiated with X-rays as derived from panel (**D**). Each experiment was performed in duplicate. Each symbol in the bar graph represents an independent experiment. Data are represented as ± SD. Statistical analysis was performed by ANOVA with Bonferroni post-hoc test. *p*-values (* *p* < 0.05, ** *p* < 0.01).

**Figure 5 cells-11-03354-f005:**
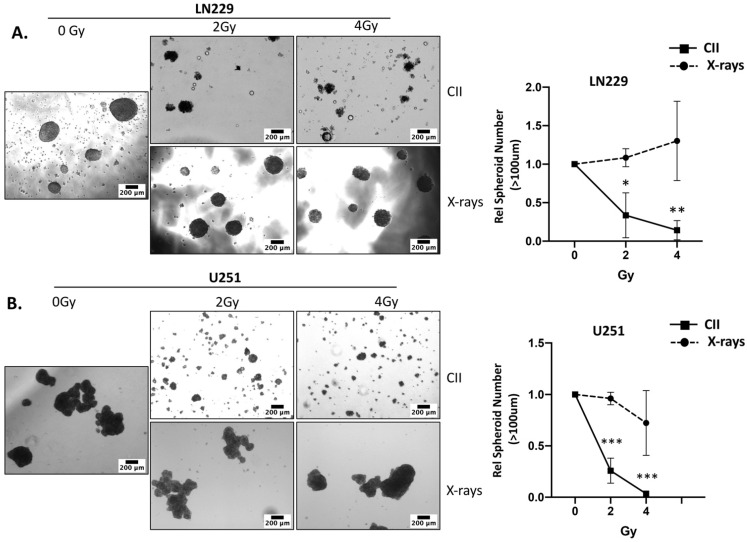
CII reduces the spheroid forming ability of LN229 and U251 glioma cell lines: Representative microscopic image of the spheroids formed by (**A**) LN229 and (**B**) U251 at day 14 post irradiation with CII or X-rays (scale bar in black = 200 μm). ((**A**), right)((**B**), right) Relative spheroid number derived from (**A**,**B**). Each experiment was carried out in triplicate. Data are from 3–4 independent experiments and presented as mean ± SD. Statistical analysis was performed by ANOVA with Bonferroni post-hoc test. *p*-values (* *p* < 0.05, ** *p* < 0.01, *** *p* < 0.001).

**Figure 6 cells-11-03354-f006:**
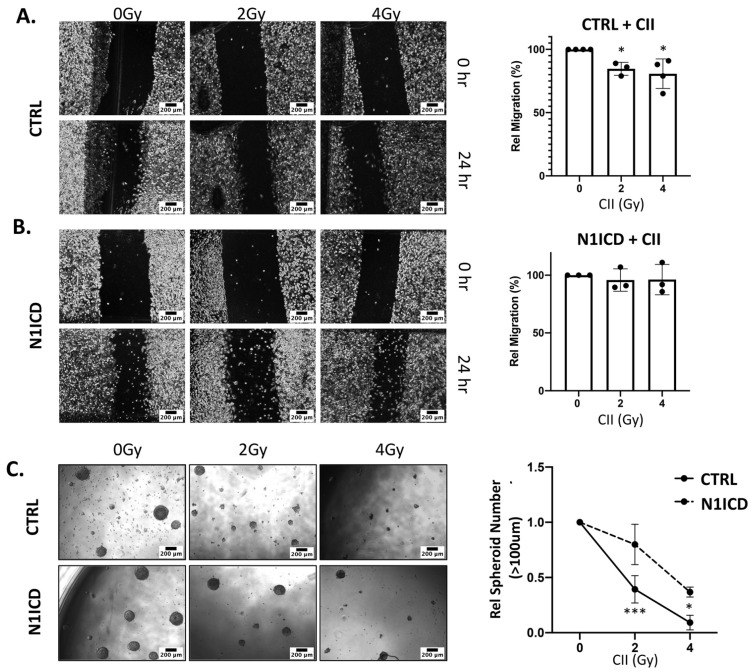
Overexpression of N1ICD attenuates the effect of CII on LN229 migration and spheroid formation: Representative images of the cell migration after CII irradiation on ((**A**), left) Control or ((**B**), left) N1ICD overexpressing LN229 cells, as analyzed by scratch assay (scale bar in black = 200 μm); ((**A**), right) Relative migration derived from panel (**A**); ((**B**), right) Relative migration derived from panel (**B**). Each symbol in the bar graph represents an independent experiment. Data are presented as mean ± SD. *p* values were determined by an unpaired two-tailed Student’s *t*-test, * *p* < 0.05; ((**C**), left) Representative microscopic image of the spheroid was formed at day 14 (scale bar in black = 200 μm); ((**C**), right) Relative spheroid number derived from (**C**). Each experiment was performed in triplicate. Data are from 3–4 independent experiments and presented as mean ± SD. Statistical analysis was performed by ANOVA with Bonferroni post-hoc test. *p*-values (* *p* < 0.05, *** *p* < 0.001).

**Figure 7 cells-11-03354-f007:**
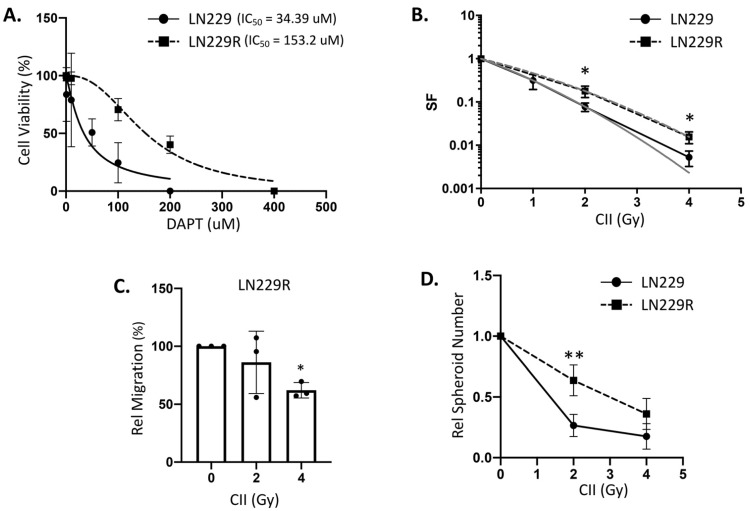
Notch inhibitor (DAPT) resistant glioma cell line (LN229R) is less responsive to CII-mediated effects: (**A**) Cell viability of parental LN229 vs. DAPT resistant LN229R cell lines as measured by Cell Titre Lumi after 72 h incubation in different concentration of DAPT; (**B**) Clonogenic survival curve of LN229 (solid black line) and LN229R (dashed black line) glioma cell lines exposed to different doses of CII, the grey line represents the linear quadratic (LQ) fitted survival curve; (**C**) Relative migration of LN229R cell line at 24 h exposed to different doses of CII. Each symbol represents one independent experiment. Each experiment was performed in duplicate; (**D**) Relative spheroid number of LN229R glioma cell line at day 10–14 after irradiation with CII. Each experiment was performed in triplicate. Data are from 3–4 independent experiments and presented as mean ± SD. Statistical analysis was performed by ANOVA with Bonferroni post-hoc test. *p*-values (* *p* < 0.05, ** *p* < 0.01).

## Data Availability

Not applicable.

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
