# Peer review of "Carbon Ion Irradiation Downregulates Notch Signaling in Glioma Cell Lines, Impacting Cell Migration and Spheroid Formation"

_cells, 2022, doi:10.3390/cells11213354_

Round 1
Reviewer 1 Report
In their manuscript, Kumar et al., by investigating the effects of Carbon ion irradiation (CII) on Notch signaling, cell motility, and spheroid formation in glioma cell lines, indicate a potential role for Notch inhibition in CII-mediated antitumor effects. In particular, the authors demonstrated that, in contrast to X-ray, which enhanced the expression of Notch signaling components, cell migratory and spheroid forming capabilities in LN229 and U251 cells, CII did not affect or even reduce the expression of Notch pathway-related genes and the protein levels of the intracellular domains of Notch1 and Notch2, decreased the ADAM-17 activity and counteracted migration and spheroid forming ability. Furthermore, forced expression of the Notch1 intracellular domain partially attenuated the anti-migratory and anti-spheroidogenesis action of CII in LN229 cells, thus suggesting that CII anticancer effects could be due, at least in part, to Notch signaling inhibition in glioma context. Further corroborating Notch's potential implication in CII mechanisms of action, CII effects were significantly reduced in the less responsive to Notch inhibition LN229R cells if compared to the parental counterpart LN229 cells. However, future investigations will require confirming these in vitro findings in in vivo models of glioma, and translating them into clinical studies. The CII has become an area of interest in the context of a potential new therapeutic angle in glioma, and improving knowledge of molecular processes underlying CII-mediated effects may lead to new avenues for treating this brain cancer. Nonetheless, a literature review appears to indicate that the linkage of CII and the Notch repression is poorly investigated in glioma. Overall, this study is in good shape; the experiments were logically designed, and the results are quite clearly described. However, to further strengthen the manuscript and make it suitable for publication, the authors should address the following issues:
1. To further support their findings on the distinct effects of CII and Xrays on Notch signaling in the glioma cells, the authors should analyze the expression levels of the activated forms of Notch receptors after the exposure to both X-rays and CII by western blotting assays using antibodies against their gamma secretase-cleaved domains (such as Notch1Val-1744, Notch2 Val-1697). Alternatively, authors could verify the nuclear distribution of the intracellular fragments of the receptors by western blotting or immunofluorescence assays using antibodies against the C-term of the receptors.
2. (Line191). Authors should replace "CAT#D67C8" with the correct catalog number of the Notch2 antibody.
3. (Lines 297-299). Authors should reformulate the statement: "In LN229 cells, X-rays upregulated the mRNA expression of Notch receptors and target genes in a dose-dependent manner observable at 24h post-irradiation, while no significant change was observed with CII (Figure1A and 1B)", since Figures 1 shows that 24h of X-rays irradiation increased the levels of Notch1 and Hey1 genes in a dose-dependent manner, whereas no significant modulation was detectable for other Notch receptors and targets analyzed by the authors.
4. (Line 300). Consistent with Figure 1A, the authors should replace Hey1 with Hes1.
5. (Lines 302-305). Authors stated: "Similarly, in U251 X-ray induced the mRNA expression of Notch receptors and target genes in a dose-dependent manner at 6h and 24h after irradiation,". However, Figures S1C and S1D show that 6h of exposure to Xray in U251 increased the levels of all Notch receptor genes in a dose-dependent fashion, without modulating any Notch downstream genes. On the other hand, Figures 1C and 1D indicate that only Notch2 and Hes1 genes were upregulated following 24h of X-ray irradiation. Therefore, I suggest reformulating the sentence.
6. (Line 310). Consistent with Figure 2, the authors should replace 2D with 2C.
7. (Figure 1). Control samples should be mentioned.
8. (Figures 2-7 and S2). Authors should use ANOVA testing with appropriate post hoc analysis when comparing multiple groups.
9. (Figures 4A, 4D, 5, 6A, 6C, S3B and S3D). Scale bars should be added to the images.
10. (Figure S3). Hes1 is reported on figure legends but not in the figure.
11. Gene names should be written in italics, while protein names should be in capital letters (see, for example, line 300). Please revise the text and correct them.
Reviewer 2 Report
The study “Carbon ion irradiation downregulates Notch signalling in glioma cell lines, impacting cell migration and spheroid formation” by V. Kumar et al, is an interesting one. The authors have compared the effects of two different energy sources; X-ray and Carbon ion irradiation (CII) in restricting glioma cell migration and spheroid formation in order to bring improvement to the existing mode of therapy against Glioblastoma. There seems to have a better outcome to CII over X-ray in regulating glioblastoma cell migration and spheroid formation and this the authors linked to the Notch signalling, which is downregulated upon CII in contrast to its upregulation following X-ray. The authors further characterize this and observe that, while X-ray increases the expression of Notch signalling components, CII does the opposite. Also, CII downregulates ADAM17 activity whereas X-ray increases its activity and thereby the processing of Notch receptors and the intracellular Notch domains (NICD). The authors conclude that CII is a better treatment option than X-ray for glioblastoma patients. These are interesting observations and have potential to improve the overall strategy to treat glioblastoma patients.
However, there are significant concerns remain concerning the authors conclusions about regulation of Notch signalling by CII in contrast to X-ray. There is no direct evidence as to how CII does the opposite to Notch signalling components than X-ray. The results shown are rather correlative than providing direct evidences. My concerns are as mentioned below. Addressing those will significantly enhance the quality of the manuscript and make the findings relevant.
1. In Figure 1, the authors show that there is no influence of CII on various Notch and its target gene expression in both the cell lines. In fact, Notch target genes are downregulated by CII. On the other hand, there are clear upregulation of various Notch proteins and their target genes in both LN229 and U251 cell lines by X-ray. However, in the abstract and in the section 3.2, they mention that CII increased the cell surface expression of Notch proteins. How is this compatible?
2. The authors have shown in Fig. 3, a reduction in ADAM17 activity in CII treated LN229 and U251 cells and an upregulation by X-ray. At 4 Gy, CII reduced the surface expression of Notch2 on LN229 cells (lines 354-356). How? At this dose, it should have greater effect than 2 Gy and it should suppress ADAM17 activity even more, leading to an increase in cell surface expression of Notch2. What about the activity of other ADAM proteins in these situations?
3. Do the observations hold true in other Glioblastoma cell lines?
4. What is the effect of CII on Notch ligands expression, which will significantly alter the nature of Notch signalling in these cells?
5. The authors have not shown or mentioned what is the effect of CII on cell death after 6 or 24h treatment compared to X-ray irradiation. This could potentially influence the migration and sphere formation assays.
6. The authors have not provided any results or not discussed the potential benefit of CII over just using a Notch signalling inhibitor such as DAPT.
7. The authors should have tested CII plus Notch inhibitor to explore if this will have greater impact in inhibiting glioblastoma cell migration and proliferation.
8. The authors should provide some insight regarding how CII could regulate the expression of Notch signalling components.
Round 2
Reviewer 2 Report
The response of the authors to my previous round of comments is not optimal. In response to the comment #1, they have not appropriately addressed the point and also have not modified the text in the abstract and in the section 3.2. Their response to the point #2 is also not convincing. They say after 24h after 4Gy CII, they found a significant increase in the surface expression of Notch2 on LN229 cells. What about Notch1 and what is the situation in U251 cells?
Please show the similar results obtained with U87 cells as mentioned by the authors as a supplementary figure. This will make the findings of the study more credible.
As the study emphasizes on the effects of X-Ray and CII on Notch signaling, the situation with Notch ligands in their experimental setting is important to analyze and the data here will make the findings relevant.
The combination treatment of X-ray/CII plus DAPT could have been performed to investigate if additional benefit could be gained against single treatment of CII or X-ray. These could have significantly improved the validity of the findings of this study.
